# Prostate Cancer Stem Cells: The Role of CD133

**DOI:** 10.3390/cancers14215448

**Published:** 2022-11-05

**Authors:** Jianhui Yang, Omar Aljitawi, Peter Van Veldhuizen

**Affiliations:** Department of Medicine, Wilmot Cancer Institute, University of Rochester Medical Center, Rochester, NY 14642, USA

**Keywords:** prostate cancer stem cells, CD133

## Abstract

**Simple Summary:**

The current theory of the prostate cancer stem cell (PCSC) may help us understand cancer recurrence and treatment failure of prostate cancer, and CD133 is one of the most powerful biomarkers to identify and isolate PCSCs. However, the PCSC theory is still a subject of conjecture in some aspects, and the value of CD133 in PCSC or other cancer stem cells was challenged by some conflicting data. The primary aim of this review is to summarize the recent research progress of CD133 in PCSC, including its selective expression in undifferentiated cells, its correlation to treatment resistance, its gene regulation and functional analysis, and its targeted therapy in vitro, in vivo, and in clinical trials. Further elucidating the detailed mechanism of how CD133 is involved in the maintenance stemness properties, will be beneficial in developing effective PCSC-specific therapy targeting CD133.

**Abstract:**

Prostate cancer stem cells (PCSCs), possessing self-renewal properties and resistance to anticancer treatment, are possibly the leading cause of distant metastasis and treatment failure in prostate cancer (PC). CD133 is one of the most well-known and valuable cell surface markers of cancer stem cells (CSCs) in many cancers, including PC. In this article, we focus on reviewing the role of CD133 in PCSC. Any other main stem cell biomarkers in PCSC reported from key publications, as well as about vital research progress of CD133 in CSCs of different cancers, will be selectively reviewed to help us inform the main topic.

## 1. Brief Introduction of CSC and PCSC

CSCs refer to a small subset of cancer cells, theoretically, this can be even a single cancer cell, which can differentiate into a heterogeneous and hierarchy of cancer cells. Sharing a number of characteristics with normal somatic stem cells, CSCs are capable of self-renewing, asymmetric division, generation of heterogeneous lineage, differentiation into various cancer cells which make up the tumor bulk, manifesting more aggressive phenotypes and exhibiting resistance to anticancer treatment [1,2]. The existence of CSCs was first reported in acute myeloid leukemia in 1997 [3] and later in a broad spectrum of common solid tumors, including PC [4].

The existence of somatic normal stem cells is well established and widely accepted, but the CSC hypothesis in some aspects is still a subject of conjecture, such as its origin. The consensus is that the normal somatic stem cell can accumulate sequential mutations and eventually transform into a CSC. Meanwhile, accumulating evidence strongly suggests that transformed normal stem cells are not the only origin, CSCs may arise from progenitor or mature cancer cells, which re-acquire CSC abilities [5].

The primary reason for proposing the CSC hypothesis is that most metastatic cancers are clinically hard to be cured, and CSCs behave like an indestructible root of that cancer, although they are rare within the tumor bulk. According to the current CSCs theory, some inherent features of CSCs make them difficult to be eradicated by conventional anticancer therapies. Firstly, CSCs overexpress some membrane transporters and possess a more potent detoxification function [6]. Secondly, CSCs reside in a quiescent state, while most anticancer drugs target proliferating cells [7]. Thirdly, CSCs have a highly efficient ability to repair DNA damage [8]. In summary, CSCs have ancestral drug resistance.

PC is the most frequently diagnosed cancer worldwide, especially in developed countries. In the U.S., approximately 248,530 men were diagnosed with PC in 2021 [9]. Despite initial treatment success, about 30% of patients experience treatment resistance and cancer recurrence [10]. Redefined mechanisms of therapeutic resistance and clinical recurrence are needed to develop more effective therapies. The theory of PCSCs may better explain current treatment failures and pave a new road to improve clinical outcomes.

The normal prostate epithelium is composed of three main types of cells: luminal cells, basal cells and neuroendocrine (NE) cells. Luminal cells are well-differentiated and express a high level of androgen receptors (AR). Basal cells are relatively undifferentiated and express low levels of AR. NE cells are differentiated but androgen insensitive. Normal prostate stem cells (PSCs) mainly reside within the basal layer. Transformed PSCs sit at the top of the hierarchy in PC and own most features of PSCs, such as being capable of differentiation into various cancer cells. Notably, aside from transformed PSCs, PCSCs can also originate from transformed intermediate or progenitor cells which re-acquire self-renewal ability [11,12,13].

## 2. The Identification, Isolation and Enrichment of PCSC

The availability of reliable PCSC markers is essential to isolate PCSCs. Just like many other CSCs, the PCSC is likely to share similar antigen expression with PSC, its unmutated counterpart [14]. Accumulated evidence has shown that PCSCs express certain functional and non-functional (phenotypic) markers. With these markers labelled with antibodies, PCSCs can be identified by flow cytometry (FCM) and isolated by fluorescence-activated cell sorting (FACS) or magnetic cell sorting (MACS). 

Identification of PCSCs was initially reported by three independent groups in 2005 [4,15,16]. All these initial PCSC research teams have isolated tumorigenic and self-renewing cells from prostate cancer tissues or cell lines with different PCSC markers. The most influential work was from Collins et al. group [4], who isolated PCSCs from human PC biopsies with CD44^+^/α2β1^high^/CD133^+^ phenotypes. The isolated PCSCs were capable of differentiation to AR^+^/PAP^+^/CK18^+^ luminal cells. The fundamental evidence to prove the existence of PCSC is the reconstitution of a cancer bulk by inoculation of a small number of cancer cells in a xenograft model. After CD44+/α2β1high/CD133 cells were implanted subcutaneously in mice, formed acini-like structures were found to resemble prostate differentiation [4]. After that some researchers have utilized a variety of tentative PCSC markers to isolate PCSCs, or PC cells with stemness features, from patient derived tissue or PC cell lines. These markers include: CD133 [4,17], CD44 [4,18,19,20], ABCG-2 [15], CD24 [20], CD166 [21], ALDH1 [22], integrin α2β1 (CD49b) [4,18], Sca-1 [23], etc. (Table 1).

Most published papers utilized CD133 based combined markers, only a few research teams applied a single marker such as CD133 alone. In addition, the above PCSC markers can be divided into extracellular or intracellular molecules. Extracellular markers technically do not require fixation and permeabilization for antibody binding, so they are more suitable for isolating living cells, which later will be collected for downstream in vitro or in vivo experiments to analyze PCSC properties. It is noteworthy that current PCSC markers do not exclusively express on PC, most of them express in CSCs of other cancer types.

Apart from the isolation from human PC tissue, significant research work also has shown that differentiated PC cell lines can be reprogrammed to PCSCs or stem-like PC cells by exposure to a variety of harsh experiment conditions, including chemotherapy [24,25], radiotherapy [26,27], serum-free medium [24,28], low attachment culture systems [28], sphere cultures [29], and androgen deprivation [30].(Figure 1) These experimental conditions can also induce tentative CSCs or stem-like cancer cells in other types of cancers, because the induced and sorted cells are capable of constituting tumors in a xenograft mouse model. The possible mechanism for these reprogrammed CSCs is inducing de-differentiation [27] and/or selecting resistant stem-like cells, while most of the non-CSC cancer cells die in unfavorable environments. Notably, genetic manipulation, such as PTEN deletion, can also lead to the expansion of the PCSC phenotype and tumor initiation [13,31].

As mentioned earlier, the most convincing evidence of PCSC is the ability to reconstitute a tumor by the limited dilution assay, in which only a tiny amount of the tentative CSCs can demonstrate a significant ability to reconstitute the cancer bulk in severe combined immunodeficiency (SCID) mice. The Collin group showed that a few CD133+ cells, as few as 10, were able to develop a tumor [32]. In glioma cells, 100 CD133+ stem cells were sufficient to reconstitute a brain tumor [33].

## 3. CD133 Is a Robust Biomarker to Identify PCSC

In 1997, it was reported that CD133 was expressed in a subset of human CD34(+) hematopoietic stem cells (HSC) derived from human bone marrow and cord blood, so CD133 was first regarded as a marker of HSC [34,35]. In 2003, brain tumor stem cells were exclusively isolated using neural extracellular stem cell marker CD133. These CD133+ cells differentiated into tumor cells bearing some resemblance to patient-derived tumors [33]. Since then, in a variety of solid tumors, CD133 has become the most frequently used extracellular marker to detect CSCs [4,33,36,37,38,39,40,41,42,43]. Such universal expression tends to confirm CD133 as an essential maker of CSCs, despite contradicting data regarding the ambiguous role of CD133 expression in certain CSCs [36,44].

In the past two decades, CD133 alone [17,29], or in the combination with other markers [4,45,46,47,48,49], is one of the most well-characterized biomarkers used to identify PCSCs. The CD133 based PCSC marker combination includes: CD133+/CD44 [46], CD133+/CD44+/integrin α2β1 [4,45], CD133+/CXCR4 [47], CD133+/Trop-2+/integrin α2β1 [48], CD133+/CD44+/ABCG2+/CD24- [49]. It is unclear whether combined markers are more valuable than CD133 alone to identify PCSCs, but in colorectal cancer, it has been suggested that the CSC marker pool is more precise than CD133 alone [50,51]. In the first and key publication of PC, the marker combination CD44^+^/α2β1^high^/CD133^+^ was used to isolate PCSCs from 40 patient biopsies [4].

It is not surprising that CD133 expressed at low levels in prostate cancer tissues and four patient -derived PC cell lines, including PC-3, CWR22Rv1, DU-145 and LNCaP [29]. However, in the combined PCSC markers such as CD44^+^/α2β1^high^/CD133^+^, not all markers are weakly expressed in all PC tissue or cell lines. For example, PC-3 and DU-145 expressed > 93% CD44+ cells, while CWR22Rv1 and LNCaP cells expressed < 4% CD44+ cells [52]. Our unpublished data confirmed the broad and strong expression of CD44 in PC-3 or DU-145 cells by FCM and Western blot. According to the definition of a CSC, cancer stem cells are less than 1% of all cancer cells, and the expression of a CSC marker in histological slides is supposed to be weak or not even expressed in non-CSC cancer cells. However, this general rule may not apply to all cancer cell lines because a cancer cell line was initially established from one or only a few cells, and the broad expression of CD44 or other markers cannot disqualify them as PCSC markers simplistically. As a well-recognized CSC marker in PC and other cancers, CD44+ PC cells displayed significantly enhanced tumorigenicity and metastasis compared with CD44- cells [19], and small numbers of CD44+/CD24- initiated tumors in a xenograft model [20]. The CSC marker combination CD133+/CD44+, with [4,45] or without [46] integrin α2β1, may help PCSC isolation from human PC tissue compared with CD133+ alone.

In this review article, we focus on the role of CD133 as a PCSC marker because of its concrete evidence from an in vivo tumorigenicity assay, its higher specificity due to extremely low levels (<1%) in PC tissues [47,53,54] and its role of stemness maintenance in CSCs. Notably, the function of CD133 in PCSCs is largely enigmatic, but insight can be drawn from published data in CSCs of other types. After intensive research on CD133 for almost two decades, the physiological and pathological function of CD133 is still not well elucidated. Most research investigated the role of CD133 as a CSC phenotypic marker, and only limited studies explored the mechanism about how CD133 plays an essential role in the maintenance of CSC properties, which is briefly summarized as below:(1)CD133 was expressed in 7 out of 20 neuroblastoma (NB) cell lines. In the CD133 negative cell line SH-SY5Y, CD133 overexpression suppressed its cell differentiation and decreased the expression of differentiation marker proteins. In addition, it was found that CD133 regulated NB cell differentiation by suppressing RET gene transcription and in a manner dependent on p38MAPK and PI3K/Akt pathways [55].(2)The cellular and subcellular location may also help reveal the function of CD133. At the cellular level, CD133 was reported to localize at the cell membrane, especially abundant in neuroepithelial stem cells. At the subcellular level, irrespective of the cell type, CD133 was found to concentrate in the plasma membrane protrusion [56], in which it interacts with membrane cholesterol via microdomains and is involved in membrane organization by interacting with cholesterol-binding protein.(3)CD133 intracellular domains are involved in β-catenin-mediated transcriptional regulation in CSCs and for the self-renewal capability of side population cells in some selected cancer cell lines [57]. In addition, 10^4^ CD133+ U87MG cells transfected with CD133 or β-catenin shRNA did not form tumor in mice (0/6), while 10^3^ U87MG cells transfected with negative control shRNA were sufficient to form tumors in mice (6/6).

## 4. Gene Regulation and Functional Analysis of CD133 and CSC Stemness

The human CD133 (FROM1, prominin-1, AC133) gene locates on chromosome 4p15 and has 37 distinctive exons, resulting in the 12 alternatively spliced isoforms [58] of CD133 mRNA in a tissue-dependent manner. The CD133 gene transcription is regulated by five alternative promoters, three of which locate in the CpG island where DNA methylation occur. Methylation of these 133 promoters in vitro completely inhibits their activity, suggesting that methylation plays a vital role in gene regulation [59,60,61]. On the contrary, an abnormal DNA hypomethylation status of the CpG island in the promoter is positively correlated with elevated CD133 expression in some types of CSCs [59,62].

CD133 mRNA is detected in most adult tissues and in many cell lines, but CD133 protein expression is restricted and mainly expressed on normal stem cells [63,64,65,66], including prostate stem cells [65]. Only a few normal prostate cells express CD133, and most basal and luminal cells are negative, indicating CD133 expression is strictly defined during the development of epithelial hierarchy in prostate tissue. Due to heavy hypermethylation of the CpG island, DNA methylation inhibited CD133 expression in a number of prostate epithelial cell lines [67]. On the contrary, histone deacetylase inhibitors restored CD133 expression in prostate cell lines. However, in malignant prostate primary tissues, regulation of CD133 is under the dynamic control of chromatin condensation but not dependent on DNA methylation [67].

Three transcriptional factors are identified to regulate the transcription of CD133, which include the ALL1-fused gene from chromosome 4 protein (AF4), Sex determining region Y-box17(Sox17), and E26 transformation-specific (ETS). AF4 was identified to be a regulator of CD133 in Caco-2 cells (a colorectal carcinoma cell line) by shRNA screening [68]. Sox17 was identified as a critical regulator of self-renewal of fetal and adult HSC [69]. Forced expression of Sox17 induced expression of CD133 in CD133—cells, and reduction of SOX17 by siRNA induced a reduction in the level of CD133 in CD133 + cells [70]. The RAS/ERK/ETS conduct pathway was regulated at ETS binding site within the CD133 promoter [71], while suppression of the ERK pathway downregulated the expression of the CD133 protein.

CD133 is a 97kDa transmembrane glycoprotein with five transmembrane domains, and due to heavy glycosylation, its apparent molecular weight is about 130 kDa. CD133 null mice usually grow normally except for a progressive degeneration of photoreceptors [72], which is consistent with the critical function of CD133 in photoreceptor cells [73]. CD133 selectively expresses in some types of stem cells during tissue development, and its expression is regulated in a development-dependent manner. The presentation of CD133 is rapidly lost upon differentiation.

In PC, as discussed earlier, subpopulations of CD133+ cells isolated from primary prostate cancer tissues or established cell lines exhibited stem cell-like characteristics. Ectopic over-expression of CD133 rendered LnCap cells significant CSC properties such as higher expression of Oct-4 and Nanog [74], promoted bone metastasis, and increased epithelial-to-mesenchymal transition (EMT) properties, which include increased vimentin and decreased E-cadherin.

## 5. Multiple Functional Roles of CD133 and CSC Stemness

The upstream or pertaining molecular events inducing transcriptional, translational or epigenetic regulation of CD133 are still largely unknown. It was reported that certain conduct pathways, extracellular oxygen levels, or mitochondria metabolism were mutually interconnected with CD133 gene expression and stemness of CSCs, as summarized below and illustrated in Figure 2:(1)PI3K/Akt pathway: In glioma CSCs, phosphorylation of tyrosine-828 in the CD133 C-terminal domain mediated interaction between CD133 and the phosphoinositide PI3K 85 kDa subunit (p85), which further activated the PI3K/Akt conduct pathway. On the contrary, CD133 knockdown significantly inhibited the activation of the PI3K/Akt pathway, accompanied by reduced properties of self-renewal and tumor-forming in glioma CSCs. Taken together, CD133 activated the PI3K/Akt pathway and regulated stemness in glioma CSCs [12,75].(2)Wnt Signaling: In several patient-derived glioblastoma cell lines, compared with CD133 ^low^ cells, CD133 ^high^ cells showed higher levels of endogenous Wnt activity and self-renewal property, while inhibition of CD 133 by a novel anti CD133 antibody suppressed the function of CD133 as well as the activity of Wnt pathway. Interestingly, a pan-AKT inhibitor MK-2006 diminished overexpression of CD133 induced Wnt activation, indicating a CD133/AKT/Wnt signaling axis may play a role in regulating the stemness of glioblastoma [76].In PC, non-adherent prostaspheres cultures enriched stemness characteristics of prostate cell likes. Inhibition of Wnt signaling reduced the prostasphere size and the self-renewal properties of prostate cancer stem-like cells, while adding Wnt3α increased self-renewal and expression level of CD133 [77]. Therefore, Wnt-β-catenin signals promote the self-renewal of PCSC or progenitor cells [78], which may be independent of AR activity [77].(3)CD133-transferrin-iron: The low oxygen niche is the microenvironment where the stem cell resides. In the tumor microenvironment, hypoxia upregulated the expression of hypoxia -inducible factor-1(HIF-1) and then indirectly induced CD133 expression [79,80] and other stem cell markers of PCSC [81]. In addition, hypoxia also disturbs mitochondrial membrane potential (MMP) to regulate CD133 post-transcriptionally [82].(4)Reactive oxygen species (ROS): ROS are by-products of normal cellular metabolism but excess ROS leads to cell death. In CSCs, the Redox scavenger system is activated to keep ROS at a low level [83]. In PCSCs, CD133+ cells are more vulnerable to ROS-induced cell damage [84].

## 6. CD133 and Its Clinical Significance in PCSC

It is estimated that metastasis is responsible for about 90% of cancer deaths. [85]. An autopsy study showed that bone metastases were found in 90% of 1589 patients who died from metastatic prostate cancer, strongly suggesting this preponderance of bone metastasis in castrate resistance PC [86]. It is likely that bone metastasis is the ultimate result of PCSC disseminated from prostate cancer [87].

A cancer metastasis initiates from an invasion of cancer cells through the basement membrane, followed by multiple steps including angiogenesis, intravasation, extravasation, and colonization. In addition, cancer metastasis requires an epithelial status switch, including both an EMT [88] to leave the primary location and a Mesenchymal-to-Epithelial Transition (MET) to seed into the secondary site [89]. PCSCs are more apt to an epithelial status switch and metastasis because of the capacity of cell plasticity. CD133 may be involved in cancer metastasis, especially bone metastasis. In a group of 131 cancer patients (26% prostate cancers), 111 metastatic patients had a significantly increased expression of CD133 mRNA (*p* < 0.05), especially patients with bone metastasis (*p* < 0.001) [90].

It is reasonable to postulate that CD133 might be a progress factor in some solid cancers, and several reports correlated the expression of CD133 with poor prognosis in a variety of solid cancers [91,92,93,94,95]. In a group of metastatic castration-resistant prostate cancer (mCRPC) patients, circulating tumor cells (CTCs) with CD133+ have an AR-independent, increased proliferative potential [96]. Very recently, a clinical trial was conducted to evaluate the clinical significance of CD133 in CTCs of newly diagnosed mCRPC patients. It was found that using CD133 in circulating tumor cells can independently predict progression-free survival (PFS) in mCRPC patients who received androgen deprivation therapy (ADT) therapy (*p* < 0.05) [97].

Extracellular vesicles (EVs) are secreted by cells into the extracellular space and play important roles in cell–cell communication. Blood circulating EVs released from cancer cells or CSCs may become potential biomarkers for cancer diagnosis and prognosis prediction [98,99]. It was reported that in stage IV colorectal cancer patients, total EVs and CD133+ EVs in blood before treatment were significantly associated with poor survival and reduced treatment response [100]. Currently, the prognostic and predictive value of CD133 EVs in PC has not been reported and still needs investigation.

## 7. CD133 and Drug Resistance

Normal stem cells express higher levels of chemotherapy-resistant proteins such as P-glycoprotein [101] and ATP binding cassette subfamily G member (ABCG2) [102], so they are more resistant to anticancer treatment by nature, regardless of their mutagenesis or tumorigenesis status. However, whether CSC markers can be directly applied to monitor or predict therapeutic effectiveness still needs more evidence.

When analyzing gene profiles of 80 glioblastomas, CD133 emerged as a predictor for drug resistance and treatment outcome in patients treated with concurrent chemoradiotherapy (*n* = 42; *p* = 0.004) [103]. In PC, the expression levels of CD133 in PC cell lines are increased by ADT [104] or irradiation (single or multiple) [105], indicating a potential role of CD133 in the treatment resistance.

In PC, CD133 silencing in combination with paclitaxel synergistically suppressed cell migration and proliferation and increased the chemosensitivity compared with each treatment alone [106]. Given the lack of efficacious treatment for CRPC, Tan et al. [107] developed a CD133 antibody -based multifunctional nanoplatform, combining photothermal therapy, photodynamic therapy and chemotherapy. Results have shown that the nanoplatform achieves synergistic antitumor effects on mCRPC in vitro and in vivo [107].

In a head and neck squamous cell carcinoma (HNSCC) cell line, excessive expression of CD133 conferred chemoresistance to cell death treated by 5-FU or cisplatin through increased stemness [108]. In addition, overexpression of CD133 in rat C6 glioma cells manifested chemoresistance to camptothecin and doxorubicin by upregulating the transcription activity of p-glycoprotein-1.

FACS Sorted CD133(+) cancer cells demonstrated significant treatment resistance to chemotherapy and radiotherapy when compared with matched CD133(−) counterparts [76,109,110]. A signaling axis including PI3K/AKT/NF-κB/anti-apoptosis genes (Figure 3) is proposed and may explain partly the mechanism of treatment resistance in CD133(+) cells. The hypothetic mechanism is based on the following findings: (1) As mentioned earlier, the CD133 molecule contains a short C-terminal cytoplasmic domain with five tyrosine residues, the phosphorylation of tyrosine-828 residue mediates direct interaction with regulatory subunit (p85) of phosphoinositide 3-kinase (PI3K) [75,111]. (2) One of the main downstream effectors of PI3K is AKT. It was found that sorted CD133^high^ HCC cells expressed a higher level of active AKT (phosphorylated at Ser473), but there were no elevated MEK/ERK signaling cascade proteins [109]. (3) Elevation of phosphorylated NF-κB/p65 in whole cells and NF-κB/p65 in the nucleus were found after transfection with CD133 overexpression plasmid compared with empty vector [76]. (4) CD133(+) cells expressed a higher level of multiple anti-apoptosis genes including Bcl-2 and some members of the Inhibitors of Apoptosis Proteins (IAPs) family [110]. Besides, multidrug resistance 1 (MDR-1) was also found to be remarkedly elevated after CD133 overexpression, while CD133 knockdown plasmid decreased MDR1 expression [112].

## 8. Targetting PCSC by CD133

It is crucially important to develop PCSC-specific treatments to improve the clinical outcome of mCRPC, and these approaches [105] include the inhibition of PCSC specific signaling pathways, induction of differentiation of PCSC followed by ADT, targeting the alteration of PCSC metabolism and immunotherapy targeting PCSC markers. One of the main concerns of the potential side effects is also negatively affecting normal stem cells in the prostate, but fortunately, the adverse effects might be a minor problem because the prostate is not a vital organ to maintain normal health.

Considering the resistance of CSCs toward chemotherapy and radiotherapy, researchers explored the possibility of immunotherapy targeting extracellular antigens on PCSCs, such as the epithelial cell adhesion molecule (EpCAM). Results have shown that this approach can significantly stop the growth of PC-3 cells with highly expressed EpCAM [113]. The cell surface markers CD133 might be another target for immunotherapy or other strategies to treat PC or other cancers containing CSCs.

A novel targeted toxin, named “dCD133KDE”, was synthesized using anti-CD133 scFC which recognizing both un-glycosylated and glycosylated forms of CD133. dCD133KDE can selectively inhibit CD133 expressing subpopulations of two squamous carcinoma cell lines in vitro. When implanted into nude mice, CD133 sorted cells grew much faster and more significantly than unsorted cells. However, implantation pretreated with dCD133KDEL showed the slowest tumor formation [114].

CSCs are likely to be potential targets for immune therapeutics such as CAR T cells. CD133 is one of the main CSC markers of glioblastoma multiforme (GBM), a fast-growing and aggressive brain tumor. AC133 (epitope of CD133)-specific chimeric antigen receptor (CAR) was constructed and the AC133-CAR T cells recognized CSCs of GBM. Moreover, the AC133-specific CAR T cells inhibited the growth of orthotopic xenografts initiated from CSCs of GBM [115]. In recent phase I clinical trial for CART-CD133 therapy [116], 23 patients were enrolled, including 14 hepatocellular carcinomas (HCC), 7 pancreatic carcinomas, and 2 colorectal carcinomas. No severe cytotoxicity was observed among these patients, and the major side effect was a decreased level of hemoglobin and platelet count which was self-limited within a week. Three of 23 patients experienced a partial response, and 14 of 23 patients achieved stable disease. Tissues biopsy and immunohistochemistry confirmed that CD133+ cells were eliminated after CART-CD133 treatments.

Aptamers are single-stranded oligonucleotides that can bind to specific proteins with high affinity and low immunogenicity. In addition, aptamers have other advantages over antibody constructs including easier generation, less costly, and improved tissue penetration. Two CD133 RNA Aptamers (A15 and B19) were screened to bind to the AC133 epitope of CD 133 specifically and had greater penetration, retention and internalization in colon cancer in vitro 3D culture [117]. In prostate cancer, curcumin (CUR)-loaded CD133 aptamer A15 decreased the volume of transplanted DU-145 cells in mice compared with control groups [118].

## 9. Conclusions

mCRPC and bone metastasis are clinically tricky problems. The PCSC concept brings new horizons to develop innovative treatments to overcome these problems. A lot of approaches with potential clinical significance have been explored, and studies of PCSCs in the past two decades have gained remarkable progress.

Despite of some contradictory data, a large number of studies strongly support PCSC as the root of PC and CD133 as a key marker of PCSC stemness. In addition to known molecular events related to CD133 as mentioned above and summarized below (Figure 2), many fundamental questions still remain to be answered, with the two following questions being chosen as essential and critical: What is the pathologic and physiological function of CD133? How is CD133 involved in the signaling pathways of stemness maintenance?

The ultimate test to assess the value of the PCSC research is to improve disease management of mCRPC [5]. Understanding the regulatory role of CD133 in PCSC stemness is pivotal to the development of some effective approaches exclusively targeting PCSCs, the root of PC.

## Figures and Tables

**Figure 1 cancers-14-05448-f001:**
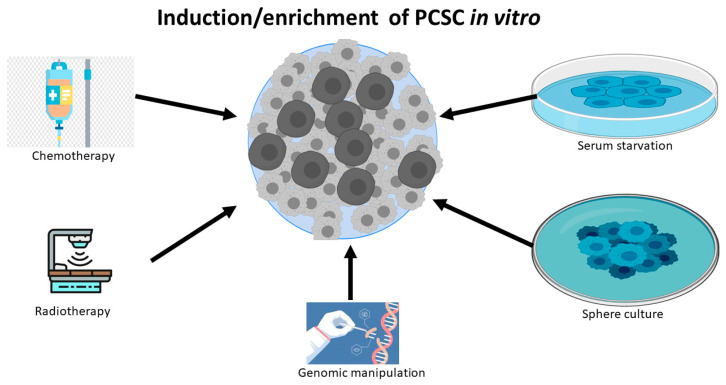
PC cell lines can be de-differentiated to PCSCs or stem-like PC cells by chemotherapy, radiotherapy, serum starvation, sphere culture, and genomic manipulation.

**Figure 2 cancers-14-05448-f002:**
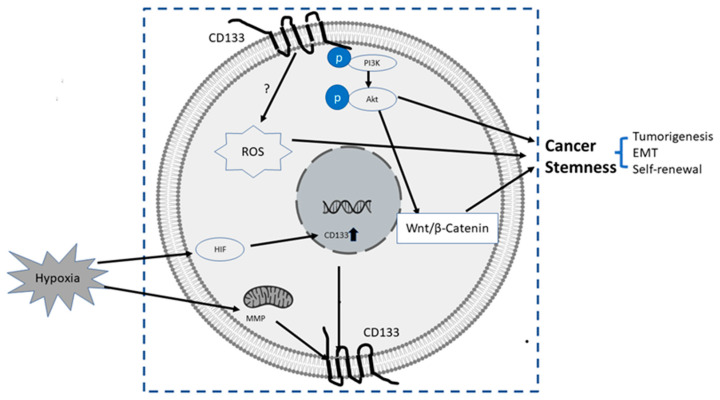
Summary of some known upstream and downstream molecular events related to CD133. The upstream events result in the upregulation of CD 133, and downstream events after CD133 induction are directly or indirectly related to increased stemness properties of PC cells, mainly via CD133/PI3K/AKT/Wnt/β-Catenin signaling axis.

**Figure 3 cancers-14-05448-f003:**
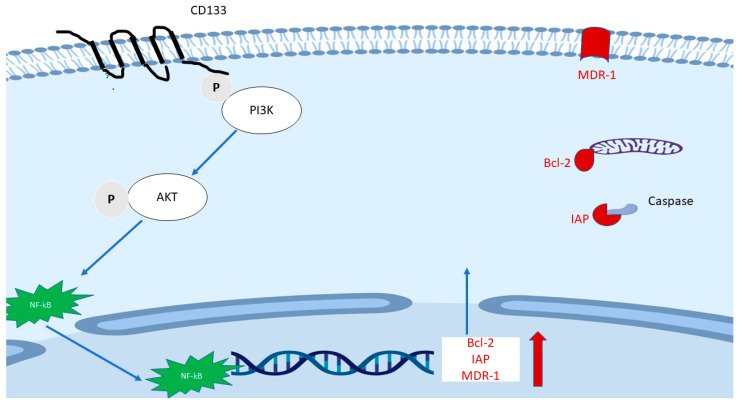
The proposed main mechanism of the role of CD133 in treatment resistance. The phosphorylated tyrosine 828 of CD133 mediates direct interaction with p85 of phosphoinositide 3-kinase (PI3K), followed by phosphorylated AK and translocation of NF-κB/p65. The latter enhances the transcription of Bcl-2, some members of the Inhibitors of Apoptosis Proteins (IAPs) family and multidrug resistance 1 (MDR-1).

**Table 1 cancers-14-05448-t001:** Current Putative PCSC markers.

Marker	Characteristic Other than as a Putative PCSC Marker	Reference
**CD133**	CD133 is a five-transmembrane domain glycoprotein localizes to membrane protrusions.	[4,16]
**CD44**	CD44 is a multifunctional surface glycoprotein involved in cell signaling, migration and homing.	[17,18]
**CD49b**	CD49b plays a critical role in both cell adhesion and lymphocyte activation.	[17]
**ABCG-2**	ABCG-2 contributes to the resistance to chemotherapeutic drugs.	[14]
**CD24**	CD24 is a cell adhesion molecule involved in the regulation of B-cell proliferation and maturation.	[19]
**CD166**	CD166 mediates cell-cell adhesion and also plays a role in development and neutrophil migration.	[20]
**ALDH-1**	ALDH1 is involved in alcohol metabolism and retinoid signaling pathway.	[21]

## Data Availability

No applicable.

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
