# Peer review of "Prostate Cancer Stem Cells: The Role of CD133"

_cancers, 2022, doi:10.3390/cancers14215448_

Round 1

Reviewer 1 Report

This review describes the role of CD133 in prostate cancer stem cells. The review is of some interest. However, I do not think it is up to the standard of being published.

1. The illustrations are too crude, and I suggest that the author redraw the illustrations.

2. There are no apparently original findings in the content of this review.

3. In part “5. Multiple functional roles of CD133 and CSC stemness”, the authors mention that CD133 is linked to several signaling pathways such as Wnt signaling and PI3K/AKT, however, their relationship is not clearly articulated. In this section, it would be easier for the reader to understand if an illustration was drawn.

4. In section “6.CD133 and its clinical significance in PCSC and CSC of other cancers”, it would have been better if the authors had concentrated on the clinical significance of CD133 in prostate cancer stem cells; this section mentions that the clinical significance of CD133 in other cancers is not relevant to this review, but rather scatters the structure of the review.

5. In Part “7.CD133 and drug resistance”, the authors mention the relationship between CD133 and drug resistance, which should then elucidate the mechanisms by which CD133 plays a role in drug resistance. The current manuscript does not cover this sufficiently, and it would be better if an illustration were added.

6. In the conclusion section there are no important findings or highlights that can be seen throughout the manuscript.

Reviewer 2 Report

This review article by Yang J et al. focused on the role of CD133, a well-know cell surface marker of cancer stem cells (CSCs) which is also prevalent in prostate cancer stem cells (PCSCs). The manuscript beings with a brief introduction on CSCs and PCSCs, methods of identification of CD133, its isolation and enrichment, its status as a robust, gene regulation and functional analysis of CD133 and CSC stemness, multiple functional roles of CD133 and CSC stemness, CD133’s clinical significance in PCSC, and CSC of other cancers, CD133’s role in drug resistance, how PCSC’s can be targeted by CD1333 and ends with a rationale and objective conclusion. The review should capture the interest of the wide prostate cancer and oncology audience, and it is timely. The two figures which complement the manuscript are well utilized and described appropriately.

However, a few concerns should be addressed:

1.       The first sentence in the Simple Summary should be revised for clarity.

2.       A reference(s) should be provided for the first sentence of paragraph 4 of Section 1.

3.       Section 2, paragraph 3 2nd sentence: change “In addition, the above PCSC markers can divided into ---” to “In addition, the above PCSC markers can be divided into ---”

Reviewer 3 Report

This review is generally well written. I have following suggestions to improve the quality of the manuscript:

1. Several wrong spellings were found such as a) in section 1: accumulating evidence strongly suggest should be "suggests"; b). In section 3, CXXR4 should be "CCR4", please check: c) In section 3,  Flow FCM should be "FCM".

2. In section 8, one important reference "Aptamer-CD133 targeted therapy " needs to be cited. i.e Shigdar S et al. Cancer letters, 2013;330:84-95.

3. One new section needs to be added to discuss CD133 as a biomarker in cancer prediction such as in human cancer tissues and in extracellular vesicle.

Round 2

Reviewer 1 Report

The revision of the manuscript is satisfactory.